# Improving the Method of Replacing the Defective Sections of Main Oil and Gas Pipelines Using Laser Scanning Data

Ildar Shammazov [1], Enver Dzhemilev [1,*] and Dmitry Sidorkin [2]

1    Department of Transport and Storage of Oil and Gas, Saint Petersburg Mining University, 199106 Saint Petersburg, Russia
2    Arctic Competence Center, Saint Petersburg Mining University, 199106 Saint Petersburg, Russia
*    Correspondence: enver.dzhemilev@mail.ru

**Abstract:** During the operation of main pipelines, many defects occur in the body of the pipe and on its surface. The main method for eliminating such defects is cutting out the defective section and welding a new one in its place. The cutting process is complicated by the possible sharp displacement of the ends of the pipeline located on both sides of the cutting site, which is dangerous for the lives of workers and can lead to breakage of the cutting equipment. In addition, to weld a new section, it is necessary to center the ends of the pipeline until they reach the alignment position, for which heavy, difficult-to-transport pipelayers are used, which allow centering the ends of the pipeline only by lifting them. Ensuring the possibility of such centering often requires additional digging of the repaired pipeline. Moreover, due to the large dimensions of pipelayers, payment of permits for their transportation is required. In addition, before transportation, pipelayers require their disassembly and assembly before carrying out repair work, which entails additional time and labor costs. To solve the problems described in this article, a developed design of devices for fixing and centering the ends of the pipeline is proposed, which makes it possible to fix the ends of the pipeline from their sharp displacement, and also to center them before welding a new section. A mathematical model was developed to assess the centering forces, the resulting stresses in the pipeline wall and the reaction forces that arise in the hydraulic cylinders of devices when leaving the ends of the pipeline in their sharp displacement. The initial data for the developed model are the coefficients of the polynomial describing the position of the pipeline in the repair trench. To accurately assess the position of the pipeline, a method of its laser scanning is proposed, the result of which is a point cloud of the pipeline. As part of the study, a method was also developed for the obtaining of a polynomial equation describing the bending of its central axis from a pipeline point cloud. As a result of experimental studies, this method has demonstrated sufficient accuracy in determining the position of the pipeline in the repair trench. Thus, the developed repair method makes it possible to increase both the safety of the repair and the technological and economic efficiency of the process of replacing a defective section.

**Keywords:** main pipelines; pipeline section replacement; stress–strain state; device; pipeline repair method; industrial safety; efficiency





## 1. Introduction

The sustainable development of any energy industry depends on the ability to ensure the uninterrupted operation of production facilities [1,2]. This requires ensuring the trouble-free operation of equipment installed on the territory of production [3,4]. The possibility of trouble-free operation of equipment directly depends on the technologies used and developed in the field to ensure an increase in the lifetime of the equipment operated as well as its repair with the least expenditure of time, labor and explicit costs, while also ensuring a high level of industrial safety [5,6].

One of the industries that needs the development and implementation of innovative technologies to ensure the smooth operation of equipment is the pipeline transport of oil and gas [7,8]. This area of research also requires the development of technologies that take into account the influence of oil and gas flow parameters, as well as their composition, on the operation of main pipelines [9,10].

During the operation of main pipelines, many defects requiring repair of pipelines appear in the pipe body and on its surface [11,12]. Repair methods include cutting, grinding, welding, the use of couplings of various designs and the use of nozzles with an elliptical bottom [13,14]. Nevertheless, cutting out a defective area and replacing it with a new one is a universal repair method used to eliminate any type of defects [15,16]. At the same time, this type of repair is most often used by service departments of companies [17].

The essence of the repair method with cutting out the defective section is to dig out the defective section, cut it out with the help of special pipe cutting machines, perform pipeline space hermetization, weld a new section, process welded joints and isolate and bury the repair area.

This method of repair is complicated by the elastic bending of the main pipeline. When the pipeline is cut, its ends, located at the edges of the cutting place, are sharply displaced relative to each other. This can lead to injury and death for workers, as well as tearing out of the pipe metal at the end of the cutting process.

To reduce the possible consequences of the displacement of the ends of the pipeline, the pipeline is often pressed with an excavator bucket during the repair work, which does not comply with the rules of industrial safety and regulations for the production of repair works. It is also worth noting that in order to weld a new section, it is necessary to center the ends of the pipeline relative to each other until they are aligned. For this, heavy, expensive pipelayers are used, which require additional costs for acquiring a permit for the transportation of oversized equipment [18].

In addition, speaking about the level of industrial safety, it is worth noting that in organizations involved in transportation and storage, there is one of the highest levels of industrial injuries among industries [19,20]. In particular, in the repair of main pipelines involving cutting out a defective section, the main share of fatal accidents occur precisely at the stages of making a "rough" cut of the pipeline by workers, centering its ends and welding a defect-free section. In this regard, ensuring the safety of repair work, as well as reducing the share of manual labor at these stages, makes the greatest contribution to improving the level of safety and reducing the level of injuries in the repair process as a whole [21,22].

The elastic bends of the underground pipeline, which are the cause of a sharp displacement of the ends of the pipeline, are quite often encountered along its length and are either calculated at the design stage and implemented during the construction of the pipeline or are formed both in the vertical and horizontal planes due to the characteristics of the natural environment in which the pipeline is laid [23,24]. They are also formed in the process of operation due to the fact that pipeline comes into operation with almost zero pinching with the ground after commissioning [25]. The features of the natural environment that contribute to the formation of elastic bends of the pipeline include extremely low and extremely high temperatures, high humidity, extremely unstable soils, significant elevation changes and pipelines crossing geodynamic zones [26,27]. The last of these factors is of particular danger, since it contributes to the formation of stresses critical for the metal of the pipeline wall, which leads to the largest accidents in pipeline transport [28,29].

The problem of further centering the ends of the pipeline before welding a new section with its ends is also relevant. This is due to the fact that, with a large displacement of the ends of the pipeline after cutting it, it is necessary to apply significant efforts on the part of the pipelayers to center the ends of the pipeline in the position of their alignment [30,31]. In addition, during the centering process, pipelayers do not allow the end of the pipeline to be lowered down, if necessary, but only allow it to be raised up. In these cases, additional excavation is required to lower the end of the pipeline, which significantly increases the

labor costs for repairs. This may require the use of three or more pipelayers, which reduces both technological and economic efficiency of repair work [32,33]. Pipelayers are also transported in a disassembled state, which requires their further assembly before repair work and disassembly after it are completed. It also requires additional time and labor costs. The economic efficiency of repair work using pipelayers is significantly reduced to a greater extent due to the need to pay for a permit for transporting oversized equipment to the work site, which can be up to 1000 dollars for transporting one pipelayer. For one service division of a company, this cost item can reach up to 100,000 dollars annually.

In connection with the above, the purpose of this study is to improve the technological and economic efficiency, as well as the safety of repair work, by developing a repair method that includes the steps of reliably fixing the ends of the pipeline before cutting it, as well as further centering its ends before welding a new section without the use of pipelayers.

To achieve this goal, the first step is to analyze the advantages and disadvantages of the methods proposed by specialists for the repair of main pipelines involving cutting out defective sections. Further, taking into consideration the identified advantages and disadvantages, it is required to propose a repair method that would meet the requirements of a high level of industrial safety, as well as technological and economic efficiency, due to the implementations of fixing the ends of the pipeline before cutting it and their centering without the use of pipelayers. Finally, it is necessary to verify the proposed technology based on experimental data and finite element modeling of the repair process.

## 2. Materials and Methods

Currently, while carrying out repairs involving cutting out a defective section, the following types of work are performed: opening a defective section of the main pipeline; pumping out the transported product from the repaired section of the pipeline; cutting of the defective section of the pipeline; pipeline space hermetization; centering of the ends of the pipeline; welding of a new section, quality control of weld seams and application of an anti-corrosion coating; backfilling of the pipeline with mineral soil; application of a fertile soil layer and land reclamation.

The most time-consuming stage of repair is the centering of the ends of the pipeline to the position of their alignment with the welded new section of the pipeline [34,35].

The position of alignment is characterized by a possible displacement of the edges of the pipes relative to each other up to 2 mm, and the angle of rotation between them up to 1.5° [16].

Specialists offer a number of possible repair methods and auxiliary devices to eliminate the shortcomings of the repair method used, but, nevertheless, there are none among them that would fully satisfy the above requirements.

Some experts propose to solve the problem of a sharp displacement of the ends of the pipeline by cutting it in a section with the lowest values of elastic stresses. However, this approach does not solve the problem of a sharp displacement due to the fact that cutting the pipeline in a section with the lowest elastic stresses leads to the smallest displacement value in the repaired section, which still poses a danger to workers and can lead to breakage of cutting equipment, in particular in sections with elastic bend radii less than one thousand outside diameters of the pipeline [36].

Moreover, during the repair process, specialists suggest using auxiliary devices for fixing and centering the ends of the pipeline. Some of the proposed repair methods do not imply a preliminary assessment of the loads that the devices undergo while fixing the ends of the pipeline, which, at high magnitudes, can lead to device failure [37]. In addition, it is necessary to assess the stresses that arise in the pipeline wall in the process of centering its ends, since, at high values of bending stresses, plastic deformation or destruction of the pipeline is possible. As for the designs of the proposed devices themselves, they also have a number of disadvantages. The device construction may not withstand high bending stresses arising in its elements [38]. Furthermore, not all devices allow centering of the ends of the pipeline freely and with sufficient accuracy in all directions [39].

Thus, in order to develop and implement a repair method that meets the requirements for ensuring a high level of industrial safety, as well as technological and economic efficiency, it is necessary to develop devices that can reliably fix the position of the pipeline ends and ensure their further centering. In addition, the repair method has to include a preliminary assessment of the loads that the devices undergo during a sharp displacement of the ends of the pipeline and the forces that must be applied to the ends of the pipeline to center them.

At the same time, the assessment of the loads that the devices undergo, as well as the forces to center the ends of the pipeline, should be carried out on the basis of data on the position of the pipeline directly in the repair trench.

The position of an underground pipeline during its operation is currently determined according to the data of in-line inspection [40,41]. The essence of this method is the launch of in-line inspection gauges into the pipeline, which measure the position of the pipeline underground and search for defects. However, this method does not allow objective assessment of the position of the pipeline in the repair trench, since in-line inspection is carried out at the moment when the pipeline is buried underground [42,43]. When the pipeline is dug out, its position changes due to its subsidence under its own weight, as well as the lack of resistance from the soil [44,45]. Moreover, in some sections of the main pipelines, it is impossible to ensure the passage of in-line inspection gauges due to elbows with tight bend radii and reductions or expansions greater than two inner diameters of the pipeline [46,47].

In this article, as a method for assessing the position of the pipeline in the repair trench, it is proposed to use laser scanning. The essence of this method is to use laser scanners to obtain a point cloud of a scanned pipeline [48,49]. Further, the coordinates of the pipeline surface points are transferred to a personal computer for further processing and obtaining the pipeline curved central axis. At the same time, this method makes it possible to evaluate the pipeline deflection simultaneously in both vertical and horizontal planes [50].

Thereby, the stages of this study are as follows:

The first step is to develop the method for repairing main pipelines and the design of the device for fixing and centering the ends of the pipeline based on the identified advantages and disadvantages of the currently proposed devices.

The second step is to develop a mathematical model to estimate the position in space of the curved central axis of the pipeline based on its point cloud obtained as a result of laser scanning.

The third step is to develop a mathematical model to assess the loads that the proposed devices will take during their operation, as well as the forces required to center the pipeline ends. The initial data for the mathematical model is the equation of the curved central axis of the pipeline, obtained from the results of laser scanning.

Finally, to prove the operability of the proposed method for estimating the position of the pipeline and the mathematical model, it is necessary to conduct an experimental study. The study includes laser scanning of the pipeline, obtaining the equation of its curved axis, comparing the resulting equation with the actual position of the pipeline axis in space and calculating the loads taken up by the proposed devices during their operation, as well as the forces required to center the ends of the pipeline. It is also required to check the convergence of the calculation results according to the proposed mathematical model with the results of finite element computer simulation.

## 3. Results

### 3.1. Development of a Method for Repairing Main Pipelines

The developed repair method, which allows elimination of the problem of a sharp displacement of the ends of the pipeline, as well as ensures their further centering without the use of pipelayers. The flowchart of the developed repair method is shown in Figure 1.

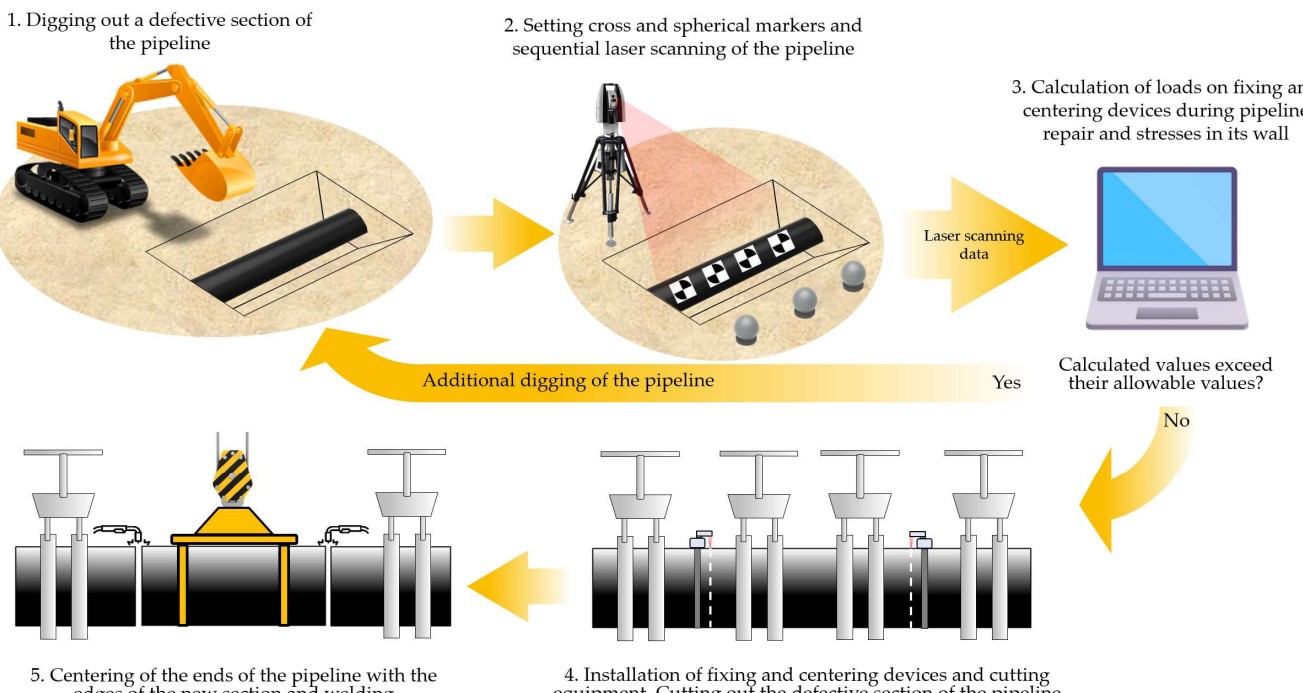

**Figure 1.** The flowchart of the developed repair method.

After the localization of the defective section of the main pipeline, for example, by the method of in-line diagnostics, it is dug out.

In accordance with Figure 2, on the upper generatrix of pipeline 1, for each of the sections in which it is planned to install devices for fixing and centering the ends of the pipeline, as well as in the sections located at the edges of the repair trench, at least one cross marker (2) is glued.

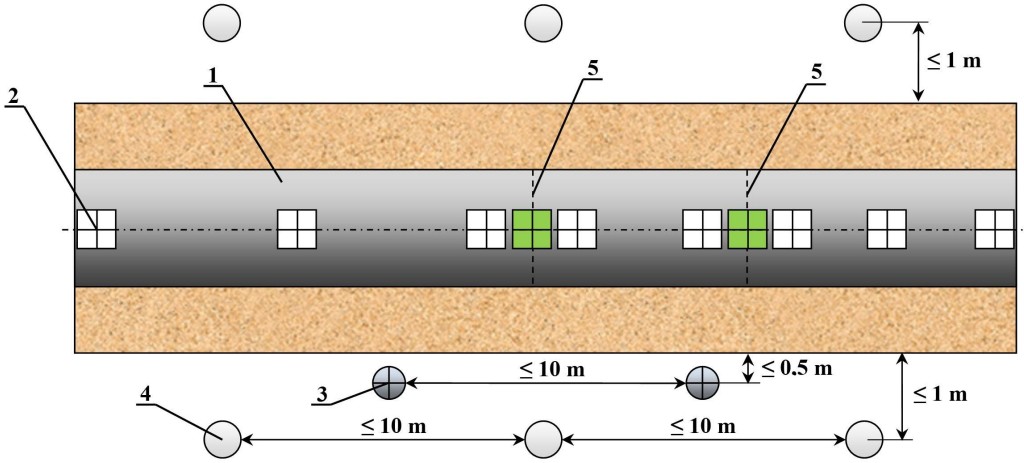

**Figure 2.** Scheme of laser scanning of a pipeline section: 1—repaired section of the main pipeline; 2—cross marker; 3—spherical marker; 4—laser scanner; 5—cutting place.

In addition, at least one cross marker (2), different in color from the rest, is glued to the upper generatrix of the pipeline in each section (5) where it is planned to cut it. At a distance of no more than 0.5 m from the edge of the trench above it on one or both sides along the length of the pipeline, spherical markers (3) are installed at a distance of no more than 10 m from each other. Next, the pipeline section is sequentially scanned along its length above the trench from one edge of the trench to the other with a step of no more than 10 m at a distance of no more than 1 m from the edge of the trench with a laser typescanner

(4), which allows scanning the position of the pipeline in space by obtaining a pipeline point cloud. After scanning the pipeline along one side, it is scanned from the opposite side. During the scanning process, the resulting point cloud is transferred to a personal computer, which performs its further processing. Merging the point clouds obtained from the scanner laser type (4) in each of its positions along the length of the pipeline is carried out by overlaying points on top of each other, which are the centers of the cross (2) and spherical (3) markers. The result of processing the point cloud of the pipeline is the equation of the polynomial describing the bending of the central axis of the pipeline.

During processing, the point cloud of the pipeline is divided by vertical planes with a step of no more than 1 m. In each plane, the points belonging to it are approximated by the ellipse equation. For each resulting ellipse, the coordinate of its center is found. Further, the points of the centers of the ellipses are approximated by a polynomial of the fourth order, which is the equation of the curved central axis of the repaired section of the pipeline.

Based on the data on the position of the pipeline in the repair trench, the calculation of the reaction forces that occur in the hydraulic cylinders of devices with a sharp displacement of the ends of the pipeline, the forces that must be created by the hydraulic cylinders to center the ends of the pipeline as well as the stresses in the pipeline wall is carried out.

If the calculated values of the reaction forces arising in the hydraulic cylinders of the clamps of the fixing and centering devices exceed the allowable values, or if the calculated stresses in the pipeline wall when centering its ends exceed the allowable stress values for the metal of the pipeline wall, the automation system notifies the operating personnel about this. After that, additional digging of the repaired section of the pipeline is required for a length of at least 10 m, and the operations for sticking cross markers in the sections of the repaired section of the pipeline and its laser scanning are carried out again.

Further, in accordance with Figures 3–5, installation of two or more fixing and centering devices, the design of which was developed in the scope of this study by authors, is carried out using a handling device on both sides of the cut-out section on the upper generatrix of the pipeline in selected sections, in which the corresponding cross markers are installed [51].

At least two fixing and centering devices are installed on the defective section of the pipeline that needs to be cut out. At the edges of each pipeline cutting place, there must be at least one fixing and centering device at a distance of no more than 0.5 m from the cutting point. After that, the pipeline is wrapped around by the grip flaps (5).

Next, installation and drilling into the ground is carried out by drilling anchors (9) using a drilling rotator installed on a hexagon. After that, the current position of the pistons of large hydraulic cylinders (3) of the fixing and centering devices is fixed by closing the hydraulic locks of the hydraulic system, thereby fixing the position of the ends of the pipeline during and after cutting.

The next step is the installation of cutting equipment (17) in the cutting places of the pipeline (13), after which the pipeline is cut. Next, the cut out defective section of the pipeline is lowered to the bottom of the trench by the fixing and centering devices installed on it, after which their drill anchors (9) and the fixing and centering devices themselves, installed on the cut out defective section (16) of the pipeline, are dismantled using a handling device. In addition, lifting from the bottom of the trench and dismantling the cut section of the pipeline are carried out with the help of a handling device.

Further, with the help of a handling device, a new section is mounted to the place of welding of its ends with the ends of the pipeline, and the pipeline ends are centered with the ends of the defect-free section by applying the previously calculated forces for centering the ends of the pipeline to large hydraulic cylinders (30 of the corresponding fixing and centering devices.

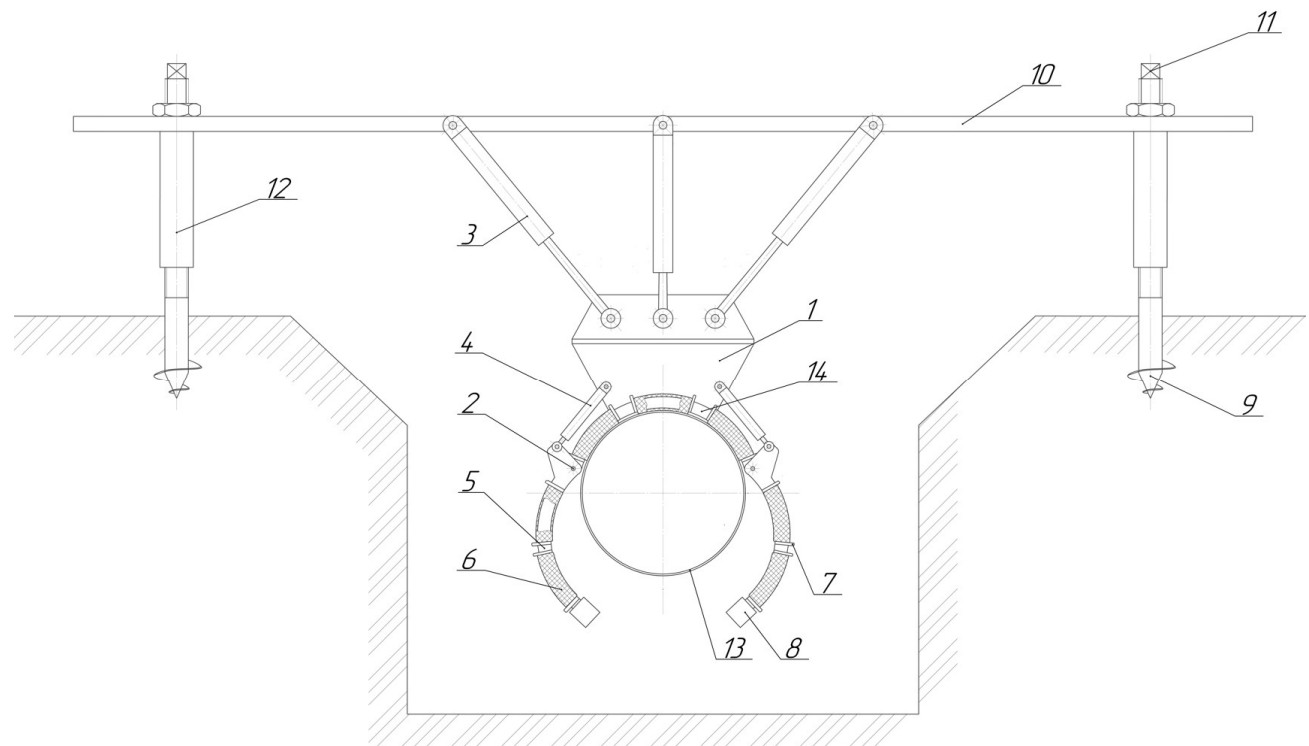

**Figure 3.** Device for fixing and centering the ends of the pipeline when cutting out its defective section: 1—the base of the grip; 2—grip axis; 3—large power cylinder; 4—small power cylinder; 5—grip flap; 6—grip roller; 7—roller stop; 8—grip lock; 9—drill anchor; 10—platform; 11—hexagon; 12—guide cylinder; 13—pipeline; 14—arc of grip.

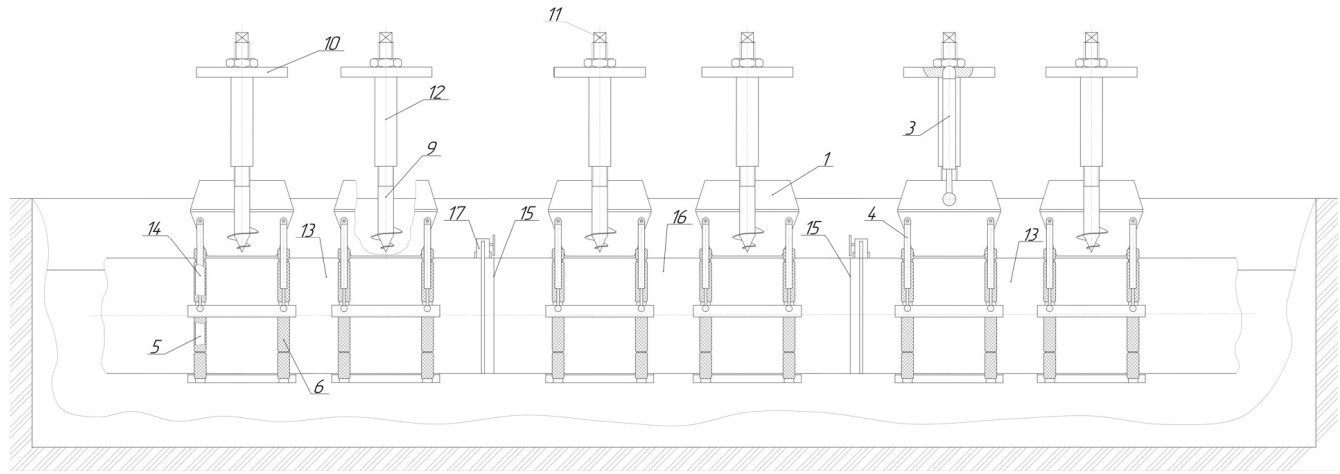

**Figure 4.** Side view of the device in the process of centering the pipeline after cutting out the defective section: 1—the base of the grip; 3—large power cylinder; 4—small power cylinder; 5—grip flap; 6—grip roller; 9—drill anchor; 10—platform; 11—hexagon; 12—guide cylinder; 13—pipeline; 14—arc of grip; 15–cutting place; 16—cut out section of the pipeline; 17—pipeline cutting machine.

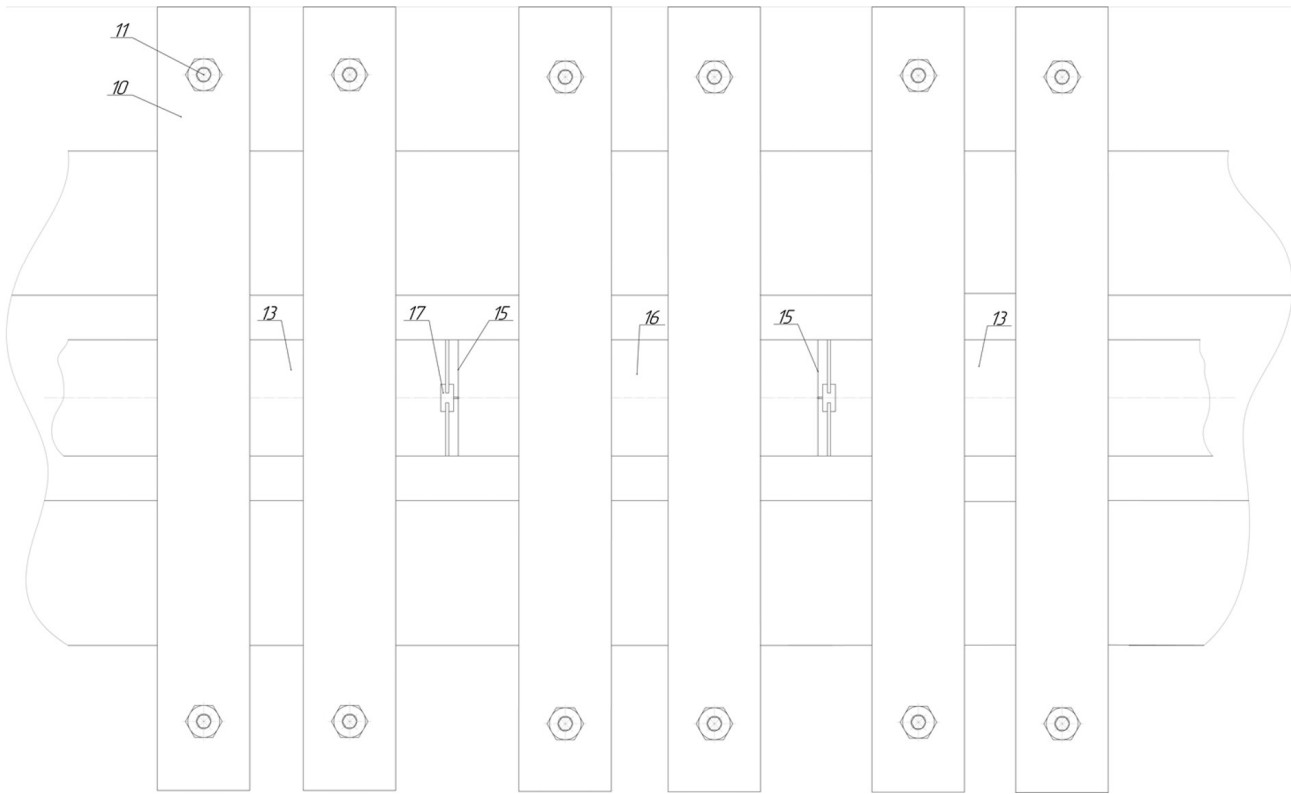

**Figure 5.** Top view of the device in the process of centering the pipeline after cutting out the defective section: 10—platform; 11—hexagon; 13—pipeline; 15—cutting place; 16—cut out section of the pipeline; 17—pipeline cutting machine.

After that, the ends of the defect-free section are welded with the ends of the pipeline, and the drill anchors (9) of the fixing and centering devices are dismantled by means of a hydraulic rotator and the fixing and centering devices themselves. Next, the quality of the welded joints is checked and their isolation, as well as the burying of the repaired section of the pipeline, is carried out.

The control of the hydraulic cylinders of the devices is carried out by the automation system by adjusting the pistons of the hydraulic cylinders in accordance with the calculated values of the forces for centering the ends of the pipeline or by remote control.

*3.2. Development of a Mathematical Model for Assessing the Loads Taken up by the Devices and the Forces Required for Centering the Ends of the Pipeline*

The initial data for the proposed mathematical model are data on the spatial position of the pipeline in the repair trench, the outer and inner diameters of the pipeline, the Young's modulus of its wall material as well as the coordinates of the location of the grips along the pipeline.

The position of the pipeline in the repair trench is characterized by two polynomials describing the bend of the central axis of the pipeline in the vertical and horizontal planes, respectively. Such polynomials have the following form [52,53]

$$y(z) = az^4 + bz^3 + cz^2 + ez \tag{1}$$

where:

*y(z)*—pipeline deflection in a vertical or horizontal plane;
*a*, *b*, *c*, *e*—polynomial coefficients;
*z*—horizontal coordinate of the pipeline section, m.

To eliminate a sharp displacement of the ends of the pipeline relative to each other when it is cut, the grips of the proposed devices are lowered by hydraulic cylinders to the upper generatrix of the pipeline, after which it is wrapped around by the grip flaps. The pistons of the hydraulic cylinders are fixed in this position and must remain motionless to keep the pipeline from deformation at the moment of possible displacement of its ends at the end of the cutting process. Such fixation of the position of the pistons is ensured by the use of hydraulic locks in the hydraulic system of the proposed devices. Hydraulic locks prevent additional inflow and outflow of the existing hydraulic fluid in the cylinders, which makes it possible to fix the position of their pistons and, as a result, fix the spatial position of the pipeline sections in which the grips are installed.

At the same time, provided that the position of the hydraulic cylinder pistons is preliminarily fixed in the position at which the pipeline is before it is cut, further deformation of the pipeline when it is cut entails a sharp increase in pressure in the hydraulic system, the maximum value of which should be taken up by it and, in particular, by hydraulic cylinders and hydraulic locks of devices.

In connection with the above, the problem of eliminating a sharp displacement of the ends of the pipeline is reduced to the problem of calculating the forces from the pipeline on the hydraulic cylinders of the devices when the pipeline is deformed at the moment of a sharp displacement of its ends.

We will solve this problem by calculating the magnitude of the reaction forces that occur in the hydraulic cylinders of the devices when the static load is suddenly removed at the place where the pipeline is cut, as a result of which the observable position of the pipeline in the repair trench is created. In this case, the pipeline oscillates around its static position (position B), in which it would be only under the action of a distributed load from its own weight with installed grips in selected places. A diagram of such pipeline vibrations is shown in Figure 6.

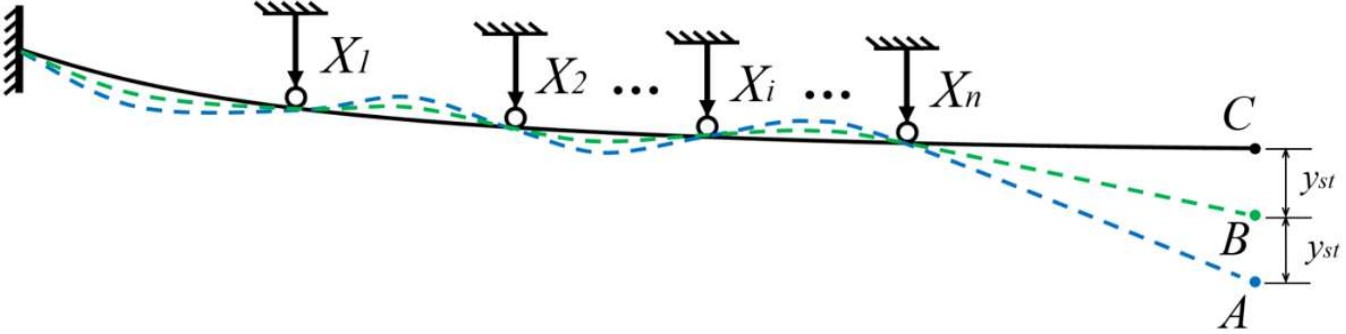

**Figure 6.** Scheme of the process of pipeline oscillations with a sudden removal of the static load at the place of its cutting: $X_i$—reaction forces arising in the hydraulic cylinders of the proposed devices as a result of pipeline deformation after cutting it, N; $y_{st}$—displacement of the end of the pipeline from its position before cutting to its position under the action of only a distributed load from its own weight on the pipeline with installed grips in selected sections, m.

Oscillations of the end of the pipeline are characterized by a change in the magnitude of the deflection at the end of the pipeline after cutting it, which changes according to the law [54]

$$y = y_{st} cos(wt) \qquad (2)$$

where:

$w$—oscillation frequency of the end of the pipeline after cutting, rad/s;
$t$—time elapsed from the beginning of oscillations, s.

The damping of oscillations can be neglected in order to avoid complicating the calculation model that will be used by the automation system that performs the calculation,

which will also provide a certain amount of the required maximum withstand force of the hydraulic cylinders and hydraulic locks of the proposed devices during their selection, since the actual values of the maximum reaction forces will be lower due to the damping of the pipeline oscillations. We also accept the model of oscillations of a beam with a concentrated mass at its end [55]. Then, the graph of the oscillation process of the end of the pipeline is shown in Figure 7.

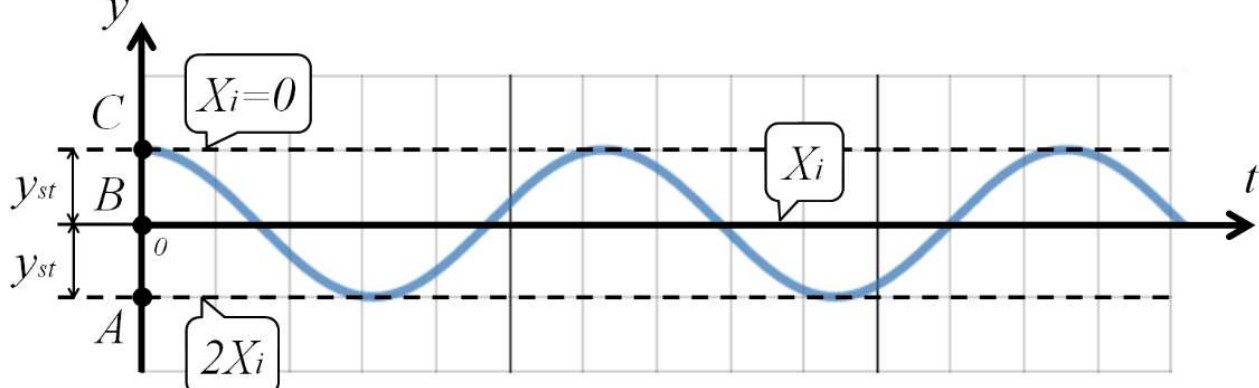

**Figure 7.** Graph of the oscillation process of the end of the pipeline.

At the initial time $t = 0$, at which the coordinate of the end of the pipeline is $y = C$, there are no reactions in the hydraulic cylinders, since in this position the pipeline is wrapped around with grippers and the position of the hydraulic cylinder pistons is fixed, while the pipeline does not experience any deformations.

The points on the graph with the coordinate $y = B = 0$ correspond to the position of the pipeline, in which it would be in a static position only under the action of a distributed load from its own weight with the location of the grips in the same positions as before cutting.

When the end of the pipeline is located at the point with the coordinate $y = A$, which is as far as possible from its initial position $y = C$ by $2 \cdot y_{st}$, the magnitude of the reactions in the hydraulic cylinders also doubles compared to the reactions in the static position (at $y = B$). These reaction values are the maximum, which the hydraulic cylinders of the proposed devices and hydraulic locks of the hydraulic system must withstand, and are equal to $X_{ic} = 2X_i$.

Thus, the problem is reduced to calculating the reactions $X_i$ that occur in hydraulic cylinders when the pipeline is deformed from the position in which it was before cutting to a static position. We will solve this problem by the force method [56].

The calculation of the reaction forces arising in the hydraulic cylinders of the fixing and centering devices with a sharp displacement of the ends of the pipeline, as well as the forces applied by the hydraulic cylinders for centering the ends of the pipeline, is carried out in accordance with the scheme shown in Figure 8.

In addition, the design scheme of the pipeline when fixing its position in the places in which the grips of the proposed devices are installed is shown in Figure 9.

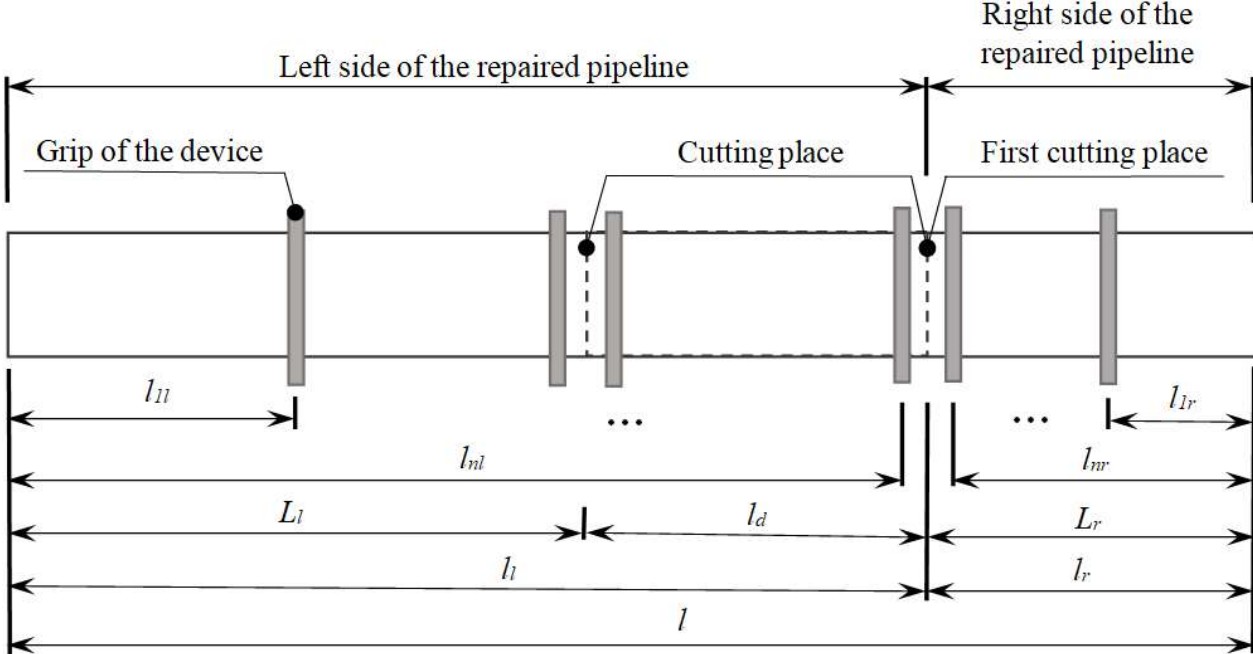

**Figure 8.** Symbols for calculating the reaction forces $X_{ic}$ according to the proposed mathematical model: $l_1$, $l_2$, $l_n$—distances from the place where the pipeline enters the pit to the places where the grips are installed, m; $l$—length of the pipeline from the wall of the trench to the cutting point, m; $l_l$, $l_r$—respectively, the length of the left and right sides of the pipeline after the first cut, m; $L$—length of the centered side of the pipeline, equal to the original length $l_l$, $l_r$ of the side of the pipeline, if the cut out section is outside it, and equal to $L$ minus $l_\partial$, if the cut out section is on this side, m; $l_\partial$—length of the cut out section of the pipeline, m.

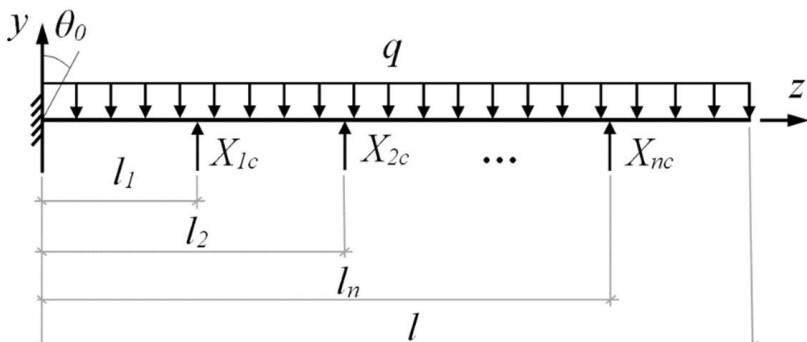

**Figure 9.** Calculation scheme of the pipeline when fixing its position in the sections in which the grips of the proposed devices are installed: $\theta_0$—angle between the axis of the pipeline and the positive direction of the OZ axis at the point of contact between the pipeline and the wall of the trench (fixed support angle), counted clockwise, rad; $q$—distributed load from the own weight of the pipeline, N/m.

The distributed load from the own weight of the pipeline (kg/m³) is calculated by the formula

$$q = \rho \cdot g \cdot \frac{\pi \cdot l \cdot (D^2 - d^2)}{4} \tag{3}$$

where:

$\rho$—density of the pipeline material, kg/m³;

$l$—length of the pipeline, m;

$g$—free-fall acceleration, m/s²;

$D, d$—outer and inner diameters of the pipeline, respectively, m.

The calculation of the forces $X_{ic}$ that occur in hydraulic cylinders with a sharp displacement of the pipeline ends when it is cut, according to the force method, is carried out by solving the following system of equations for each of the ends of the pipeline located to the left and right of the place of the first cutting of the pipeline

$$
\begin{cases}
\delta_{11}X_1 + \delta_{12}X_2 + \ldots + \delta_{1j}X_j + \ldots + \delta_{1n}X_n - \theta_0 l_1 + \Delta_{1P} = y(l_1) \\
\delta_{21}X_1 + \delta_{22}X_2 + \ldots + \delta_{2j}X_j + \ldots + \delta_{2n}X_n - \theta_0 l_2 + \Delta_{2P} = y(l_2) \\
\qquad\qquad\qquad\qquad \ldots \\
\delta_{i1}X_1 + \delta_{i2}X_2 + \ldots + \delta_{ij}X_j + \ldots + \delta_{in}X_n - \theta_0 l_i + \Delta_{iP} = y(l_i) \\
\qquad\qquad\qquad\qquad \ldots \\
\delta_{n1}X_1 + \delta_{n2}X_2 + \ldots + \delta_{nj}X_j + \ldots + \delta_{nn}X_n - \theta_0 l_n + \Delta_{nP} = y(l_n)
\end{cases}
\tag{4}
$$

where:

$\Delta_{iP}$—load terms due to the deflection of the pipeline in the cross section with the coordinate $l_i$ as a result of the action of a distributed load from its own weight, m;

$\delta_{ij}$—coefficients of the canonical equations of the force method, characterizing the displacement of the *i*-th section with the coordinate $l_i$ as a result of the action of the *j*-th force equal to unity, m/N;

$y(l_i)$—pipeline deflection in the section with coordinate $l_i$, calculated by substituting the coordinate $l_i$ into the polynomial equation, m;

*n*—amount of force applied to a given side of the pipeline with the help of grips of devices for fixing and centering its ends.

Based on the applied force method, coefficients of the canonical equations are calculated as follows

$$
\delta_{ij} = \frac{l_i^2 \left( l_j - \frac{1}{3} l_i \right)}{2EI_x}
\tag{5}
$$

where:

*E*—Young's modulus of elasticity of pipeline steel, Pa;

$I_x$—axial moment of inertia of the pipeline, m$^4$.

Axial moment of inertia of the pipeline is calculated as follows

$$
I = \frac{\pi \cdot (D^4 - d^4)}{64}
\tag{6}
$$

According to the force method, load terms are calculated by formula

$$
\Delta_{iP} = -\frac{q l_i^2}{EI_x} \cdot \left( \frac{1}{4}(l - l_i)^2 + \frac{1}{3}\left( l \cdot l_i - \frac{l_i^2}{2} \right) - \frac{l_i^2}{24} \right)
\tag{7}
$$

To calculate the forces that must be applied to the ends of the pipeline for their centering, the calculation scheme shown in Figure 10 is used.

The calculation of the forces applied by the hydraulic cylinders to lift the ends of the pipeline to the required height, ensuring the alignment of the welded pipes, is carried out on the basis of successive integration of the beam deflection equation, written in the following form [57]

$$
EI_x y''(z) = M(z)
\tag{8}
$$

where:

*y(z)*—coordinate of the vertical position of the pipeline in section *z*, m;

*M(z)*—bending moment in the pipeline cross section with coordinate *z*, N/m.

The bending moment (N·m) in the cross section of the pipeline with coordinate *z* is calculated by the formula

$$M(z) = M_0 + Q_0 z - \frac{qz^2}{2} + P_1(z - l_1) \cdot He(z - l_1) + P_2(z - l_1) \cdot He(z - l_2) + \ldots + P_n(z - l_n) \cdot He(z - l_n) \quad (9)$$

where:

$He(z - l_n)$—Heaviside function, equaled to zero for a negative value of the argument and equaled to one for its positive value.

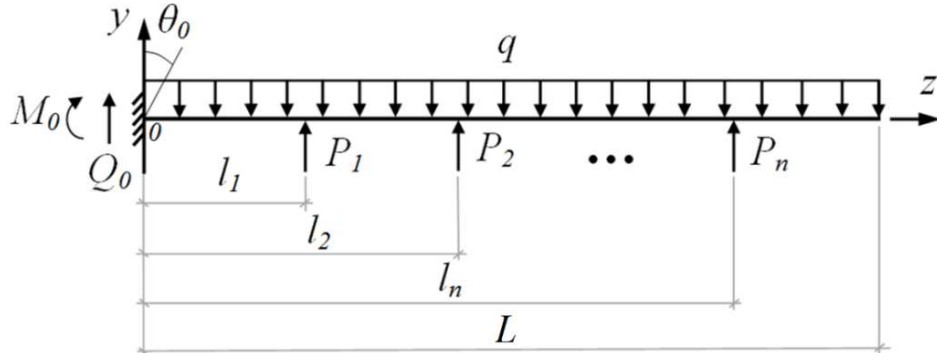

**Figure 10.** Calculation scheme for calculating the forces for centering the ends of the pipeline: $P_1$, $P_2$, $P_n$—forces applied by hydraulic cylinders, N; $Q_0$—transverse force in the place of the fixed support, N; $M_0$—bending moment in the place of the fixed support, N·m.

In this case, the boundary conditions to ensure the requirements for the alignment of the ends of the pipeline are written in the following form: $y'(0) = \theta_0$; $y(0) = 0$; $y'(l) = 0$; $y(l) = h$, where $h$—required lifting height of the end of the pipeline along the $y$ axis, m.

Then, taking into account the boundary conditions, we obtain the following system of equations

$$
\begin{cases}
\frac{P_1(L-l_1)^2}{2} + \frac{P_2(L-l_2)^2}{2} + L\frac{P_i(L-l_i)^2}{2} + L + \frac{P_n(L-l_n)^2}{2} + M_0 l + \frac{Q_0 L^2}{2} - \frac{qL^3}{6} + EI\theta_0 = 0 \\
\frac{P_1(L-l_1)^3}{6} + \frac{P_2(L-l_2)^3}{6} + L + \frac{P_i(L-l_i)^3}{6} + L + \frac{M_0 L^2}{2} + \frac{Q_0 L^3}{6} - \frac{qL^4}{24} + EI\theta_0 L - EIh = 0 \\
P_3 = 10^3 (P_1 + P_2) \\
P_s = 10^3 \left( \sum\limits_{t=1}^{s-1} P_t + \sum\limits_{w=s+1}^{n} P_w \right) \\
P_n = 10^3 \sum\limits_{u=1}^{n-1} P_u \\
Q_0 = qL - P_1 - P_2 - \ldots - P_i - \ldots - P_n \\
M_0 = -\frac{qL^2}{2} + P_1 l_1 + P_2 l_2 + \ldots + P_i l_i + \ldots + P_n l_n
\end{cases}
\quad (10)
$$

where:

$P_1$, $P_2$, $P_i$, $P_t$, $P_u$, $P_w$, $P_n$—forces applied by device grips, N; $L$—length of the centered side of the pipeline, equal to the original length of the side of the pipeline, if the cut out section is outside it, and equal to $L$ minus $l_\partial$, if the cut out section is on this side, m; $l_\partial$—length of the cut out section of the pipeline, m.

The assessment of the magnitude of the maximum stresses arising in the pipeline wall during its centering $\sigma_{max}$ (Pa) is determined as the maximum value among the stresses in the pipeline wall, calculated in each of its cross sections according to the following formula

$$\sigma(z) = \frac{M(z) \cdot D}{2I_x} \quad (11)$$

The above calculations are carried out for each of the two sides of the repaired pipeline, located to the left and right of the place of its first cutting. In this case, the considered side of the pipeline is placed with its cross section located at the edge of the trench at the origin of coordinates.

### 3.3. Experimental Study of the Possibility of Assessing the Position of the Pipeline in the Trench according to Laser Scanning Data

The following equipment was used to carry out laser scanning and verify the obtained data on the spatial position of the pipeline:

- Pipeline with a length of 2.4 m, an outer diameter of 51 mm and a wall thickness of 3 mm, all pipeline material—steel 09G2S;
- Hexagon RS6 Laser Scanner;
- Hasselblad H5D 200 MS camera with image resolution of 200 MP;
- Electrical press Testometric M350-5CT to create a load on the free end of the pipeline.

The equipment used in the process of experimental research is shown in Figures 11 and 12.

Prior to the experiment, the investigated pipeline is cleaned with a grinder in places where markers are glued for better adhesion.

Cross markers are glued along the side generatrix every 5 cm, photographed using a Hasselblad H5D 200 MS camera.

The pipeline in a bent state is shown in Figure 13.

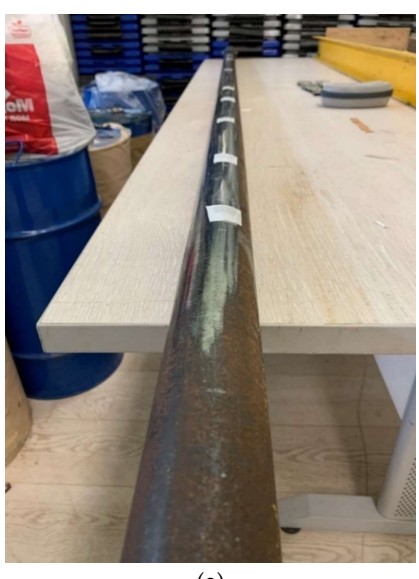
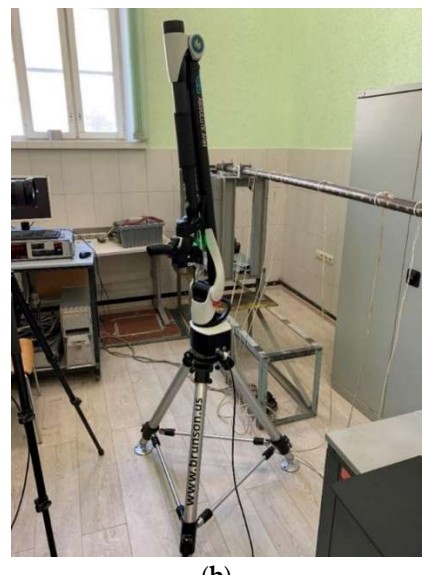
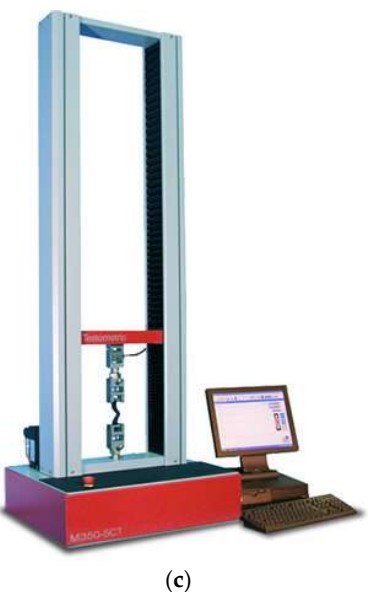

(**a**)    (**b**)    (**c**)

**Figure 11.** Laser scanner and electric press used during the experiment: (**a**) pipeline with a length of 2.4 m, an outer diameter of 51 mm and a wall thickness of 3 mm; (**b**) Hexagon RS6 Laser Scanner; (**c**) electric press Testometric M350-5CT.

The progress of the experiment is as follows:

1. The left end of the pipeline is fixed in the prepared tooling so that the length of the pipeline from the place of the fixed support is 1.6 m;
2. an electric press is installed above the second end of the pipeline;
3. a camera is installed opposite to the markers glued to the pipeline;
4. the free end of the pipeline is loaded with 400 N by lowering the electric press, thereby the elastic bending of the pipeline is realized;
5. the pipeline is photographed in a bent position using a camera;
6. laser scanning of the pipeline is carried out in a bent position with a laser scanner;
7. the load is removed from the free end of the pipeline by lifting the electric press.

The process of laser scanning of the pipeline is shown in Figure 14.

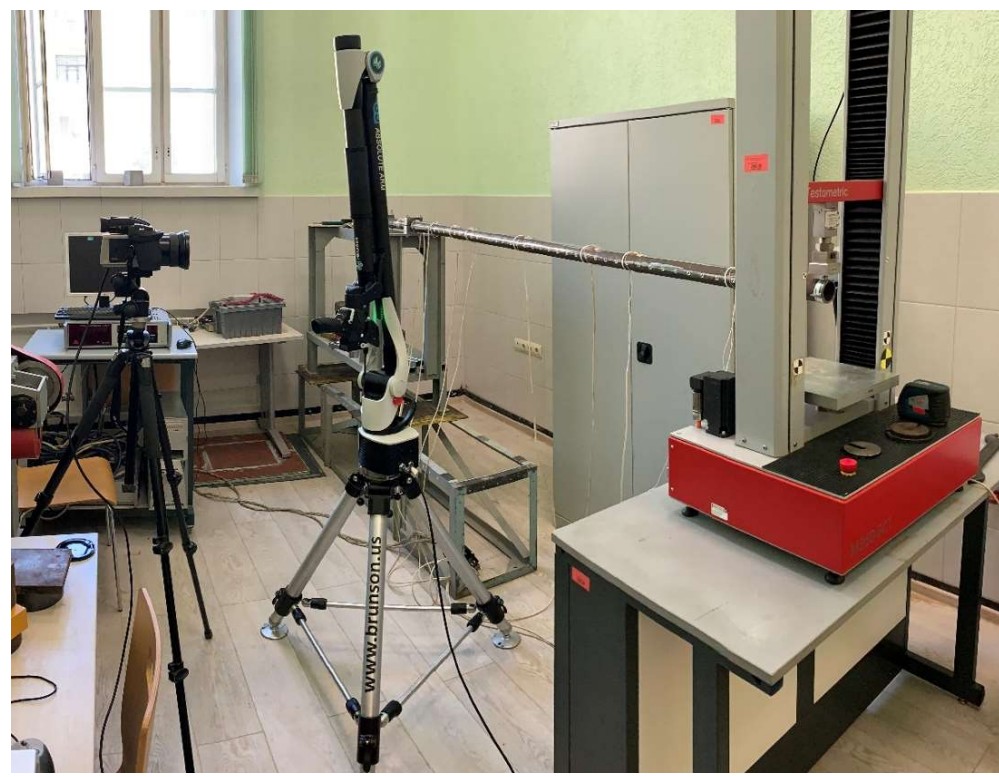

**Figure 12.** Workspace during the experiment.

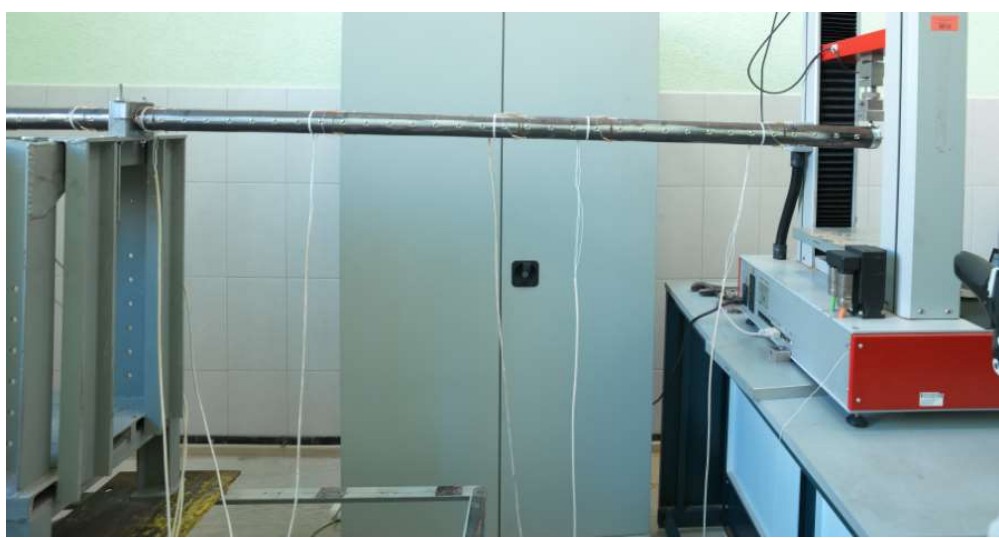

**Figure 13.** Bent pipeline.

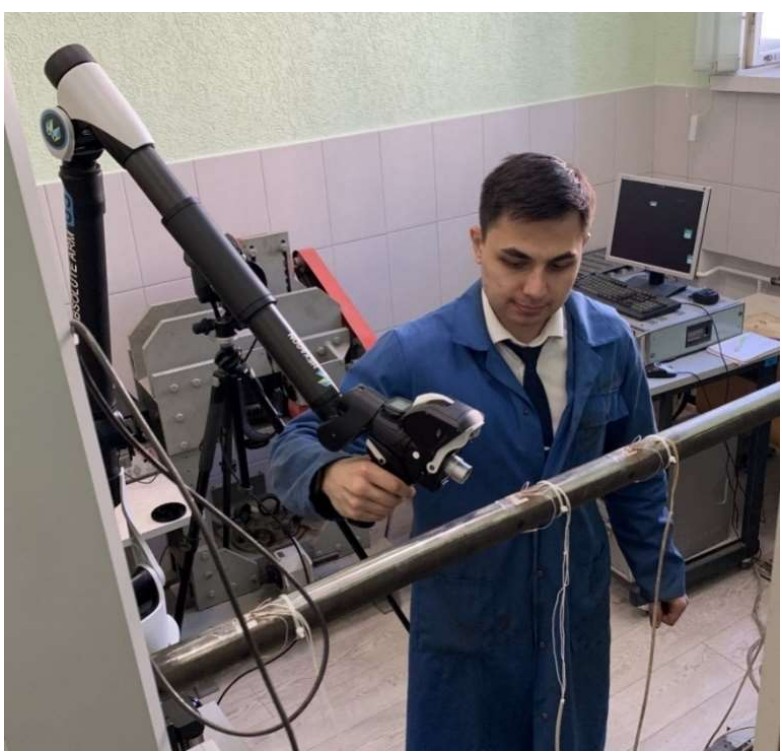

**Figure 14.** The process of laser scanning of the investigated pipeline.

The point cloud of the surface of the pipeline obtained as a result of laser scanning is shown in Figure 15.

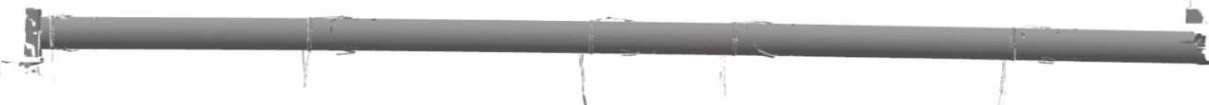

**Figure 15.** Point cloud of the scanned pipeline in a bent state.

First of all, the point cloud of the pipeline is processed, which consists of filtering the point cloud, that is, removing points with a deviation from their total set more than 0.1 mm. Points are filtered in cross sections of the pipeline free from cross markers, that is, in the middle between the markers every 5 cm. The process of filtering is carried out by highlighting the required range up to 0.1 mm on the GeometryFitGraph deviation graph shown in Figure 16. Using the SpatialAnalyzer software, many points that did not meet the requirements for the deviation value are removed.

Further, in each selected cross section, the filtered points belonging to it were approximated by an ellipse using the SpatialAnalyzer tools. Despite the fact that the pipe section is round, the ellipse equation was chosen for approximation due to the fact that when the pipeline is bent, its cross sections become ovalized [58]. For each constructed ellipse, the coordinate of the central point was found. The obtained points are the points of the curved axis of the pipeline.

Next, the bending curve of the pipeline axis is constructed based on the position of the markers.

Registration of the position of the markers is carried out with the help of a camera, for which images of the pipeline are taken when it is bent.

Next, the photos are loaded into the ThemeMotion software, the positions of the centers of the markers are indicated and the pre-measured distance between the markers is entered. For the experiment, this distance is 5 cm. Based on the distance entered, the program calculates the number of image pixels per millimeter of distance along the pipe axis.

The ThemeMotion software interface is shown in Figure 17.

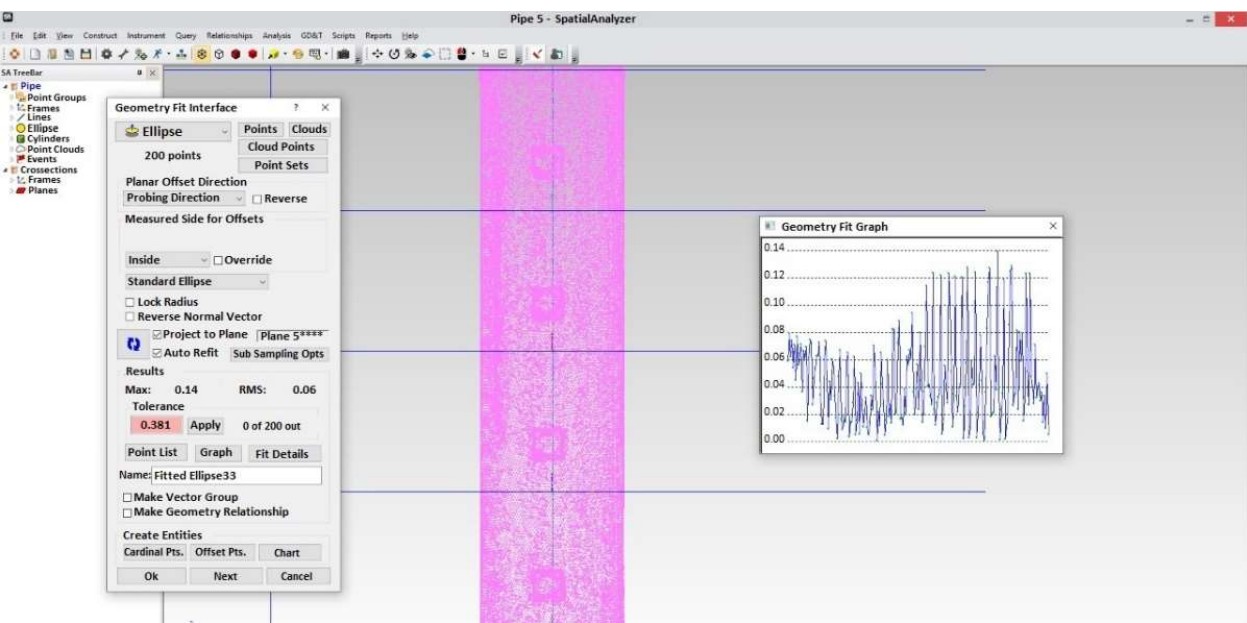

**Figure 16.** Interface of the SpatialAnalyzer software when processing a point cloud.

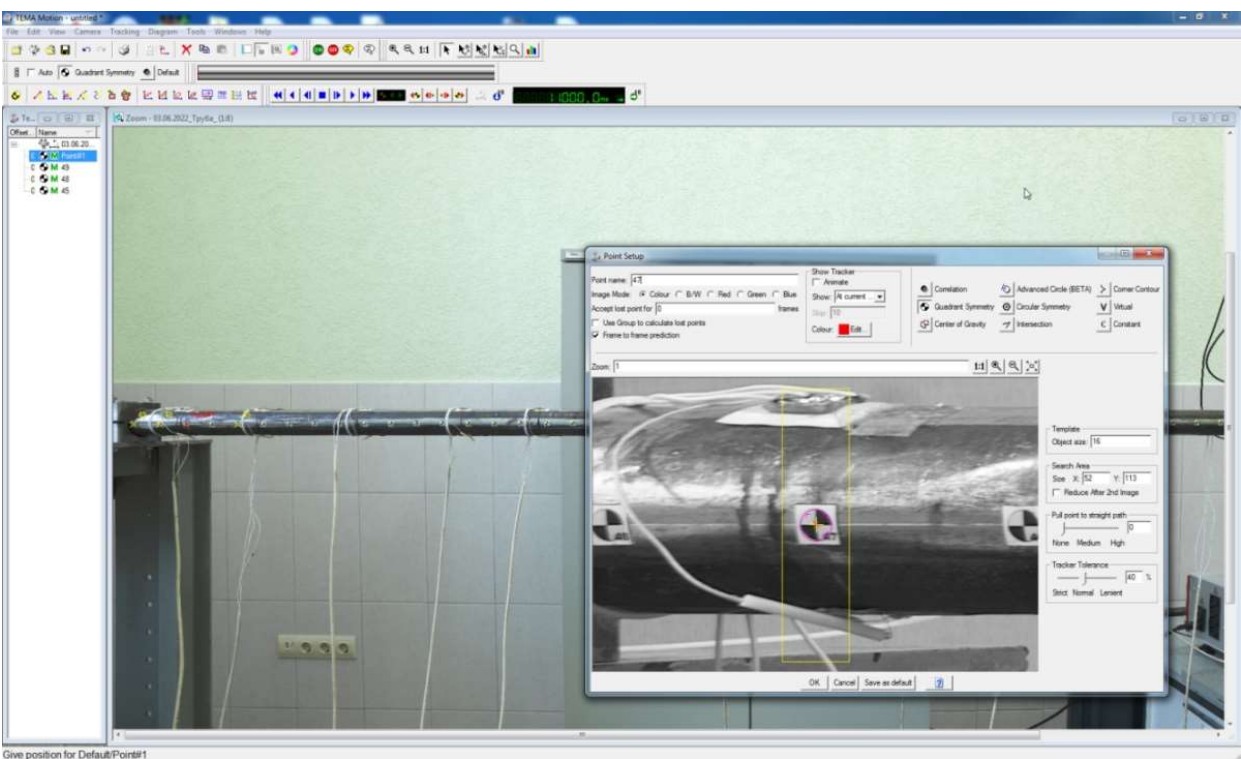

**Figure 17.** The ThemeMotion software interface.

The obtained coordinates of the marker centers are the points of the curved axis of the pipeline.

The points obtained by the two described methods were approximated by a polynomial of the fourth degree in the MS Excel with the corresponding equation display, which is a polynomial of the bent axis of the pipeline when it is bent.

The bending curves of the pipeline axis, obtained as a result of the analysis of laser scanning data and data on the position of cross markers, are shown in Figure 18.

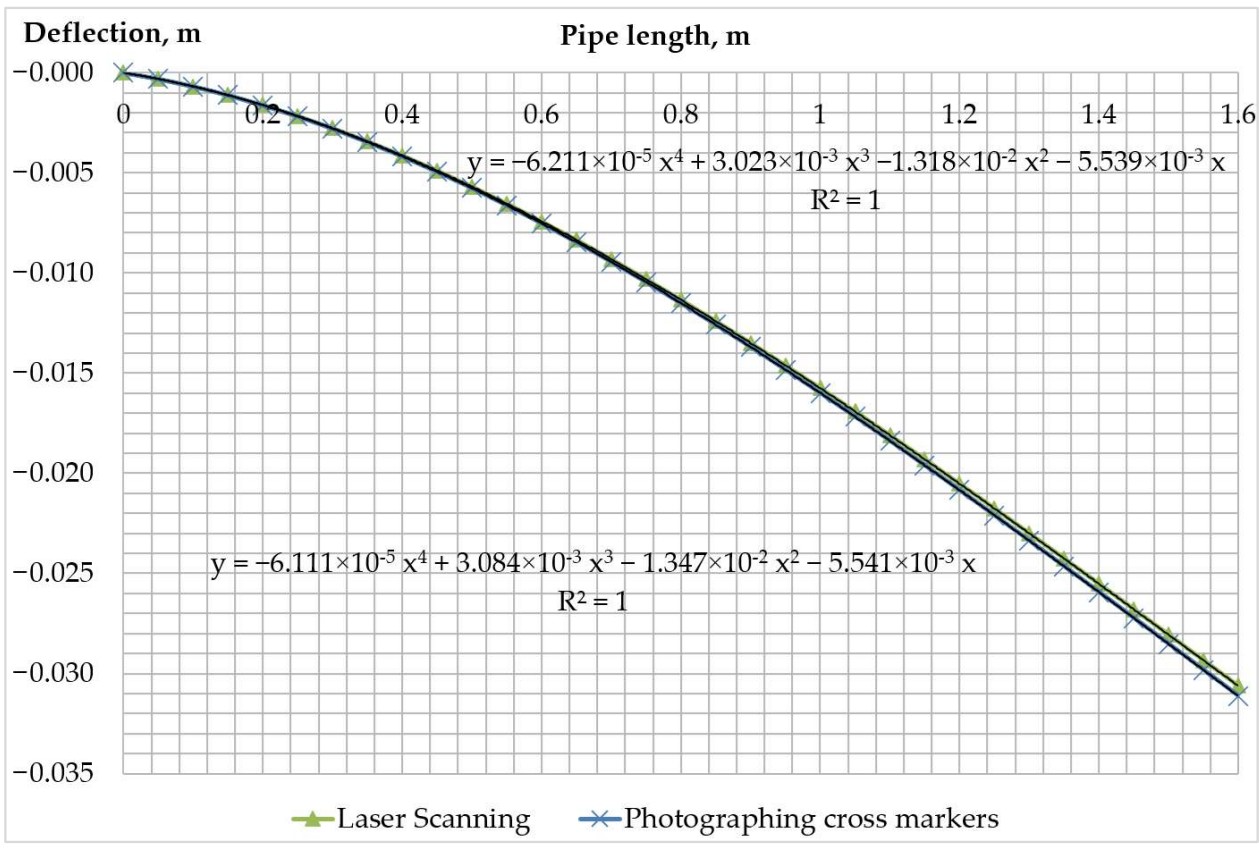

**Figure 18.** The curve of the deflection of the pipeline obtained by the method of laser scanning and photographing cross markers.

The coefficients of the polynomial for the curved axis of the pipeline in the case of its laser scanning and shooting of cross markers are shown in Table 1.

**Table 1.** Polynomial coefficients for the curved axis of the pipeline in the case of its laser scanning and photographing of markers.

| Pipeline Position Survey Method | Pipeline Axis Bending Polynomial Coefficients | | | |
|---|---|---|---|---|
| | *a* | *b* | *c* | *e* |
| Laser scanning | $-6.211 \times 10^{-5}$ | $3.023 \times 10^{-3}$ | $-1.318 \times 10^{-2}$ | $-5.539 \times 10^{-3}$ |
| Photographing cross markers | $-6.111 \times 10^{-5}$ | $3.084 \times 10^{-3}$ | $-1.347 \times 10^{-2}$ | $-5.541 \times 10^{-3}$ |

Next, the centering forces and the magnitude of the reaction forces arising in the hydraulic cylinders of the fixing and centering devices are calculated, provided that they were installed on the pipeline used during the experiment. We assume that there is a sharp removal of the load applied by the electric press from the end of the pipeline, which is equivalent to a sharp displacement of the end of the pipeline if it is cut while it is in a similar position in the repair trench. According to this, consider that the pipeline used is one of the sides of the repaired pipeline being cut. We then take the distances to the fixing and centering devices $l_1 = 1.1$ m and $l_2 = 1.5$ m. The length of the pipeline is $l = 1.6$ m. The material of the pipeline wall is steel 09G2S. The outer and inner diameters of the pipeline are $D = 0.051$ m, $d = 0.045$ m. Due to the lack of data on the position of the second end of the pipeline in the task, we take the centering height equal to $h = 0$ m. To calculate the centering forces, we use the length of the centered side $L = 1.6$ m. Based on the given initial data and the coefficients of the pipeline axis polynomials obtained by the methods of photographing cross markers and laser scanning of the pipeline, the obtained response values in the hydraulic cylinders of the fixing and centering devices, the forces required to center the pipeline end as well as the deviations of the values obtained by each of the methods relative to each other are shown in Table 2.

**Table 2.** The results of the calculation of the reaction forces arising in the hydraulic cylinders and the forces required to center the end of the pipeline used during the laboratory research.

| Parameter | Value Obtained by Laser Scanning | Value Obtained by Photographing Cross Markers | Deviations of the Values, % |
|---|---|---|---|
| $X_{1c}$, N | 125.576 | 129.152 | 2.769 |
| $X_{2c}$, N | −948.374 | −971.726 | 2.403 |
| $P_1$, N | 927.2 | 927.437 | 0.026 |
| $P_2$, N | −348.757 | −348.85 | 0.027 |
| $M_0$, N·m | 452.215 | 452.336 | 0.027 |
| $Q_0$, N | −522.731 | −522.875 | 0.028 |
| $\sigma_{max}$, MPa | 88.231 | 88.208 | 0.026 |

The calculated values of the reaction forces in the hydraulic cylinders of the devices and the centering forces obtained from the results of photographing the cross markers glued along the side generatrix of the pipeline and its laser scanning differ by less than 5%, which indicates the reliability of the laser scanning data when assessing the position of the pipeline in order to calculate the studied parameters, as well as the correctness of the above calculation algorithms.

*3.4. Determining the Convergence of the Results of Calculations Based on the Proposed Mathematical Model with the Results of Finite Element Modeling of the Process of Cutting the Main Pipeline*

In this section, we will create a finite element model of the process of cutting the main pipeline in the ANSYS software.

To do this, we build a three-dimensional model of the main pipeline with an outer diameter $D = 1.02$ m, an inner diameter $d = 0.996$ m and a length $l = 40$ m. Using the transient structural dynamic processes modeling module, we implement a bend in the pipeline, after which we create supports in the cross sections where the device grips are installed. Next, we implement the cutting of the pipeline by removing elements of the finite element mesh in the place of cutting with a width of 1 cm. After removing the mesh elements, the ends of the pipeline are displaced. At this moment, we measure the reaction forces arising in the created supports, which will be equal to the reaction forces arising in the hydraulic cylinders of the devices when fixing the position of the ends of the pipeline in the process of cutting it. The supports in this case act as an analogue of the hydraulic cylinders of the fixing and centering devices.

Next, we apply the centering forces calculated by the proposed mathematical model to the pipeline, after which we measure the height position of the ends of the pipeline.

We carry out the bending of the pipeline to an elastic bending radius equal to $1500D$, which is localized in a section with a coordinate of 25 m, in which the pipeline is cut. Centering of the ends of the pipeline is carried out at the height of the right end $h = −0.1604$ m.

The graph of the modeled pipeline deflection is shown in Figure 19.

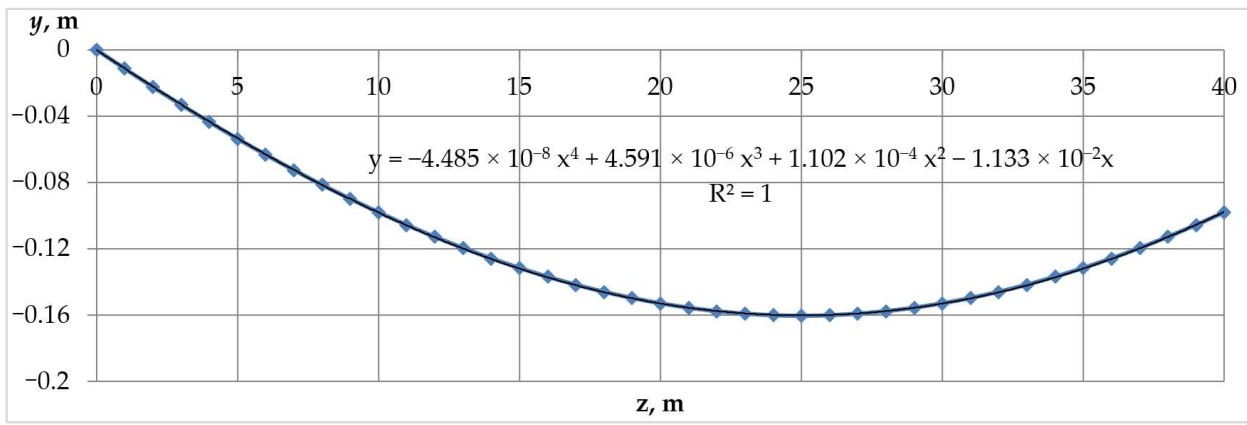

$$y = -4.485 \times 10^{-8}\, x^4 + 4.591 \times 10^{-6}\, x^3 + 1.102 \times 10^{-4}\, x^2 - 1.133 \times 10^{-2}x$$
$$R^2 = 1$$

**Figure 19.** The graph of the modeled pipeline deflection.

The initial data for the calculation are presented in Table 3.

**Table 3.** Initial data for calculation.

| Parameter | Value |
|---|---|
| Pipeline length | $l$ = 40 m |
| The length of the sides of the pipeline section after the first cut<br>–left side<br>–right side | $l_l$ = 25 m<br>$l_r$ = 15 m |
| Angle of rotation of cross sections of the pipeline located at the edge of the trench<br>–left cross section<br>–right cross section | $\theta_{0l}$ = −0.012 rad<br>$\theta_{0r}$ = −0.009 rad |
| Cut out section length | $l_d$ = 5 m |
| Distances to the grips of devices for calculating the force reactions that occur in hydraulic cylinders when fixing the ends of the pipeline and their centering forces for<br>–left side of the pipeline section<br>–right side of the pipeline section | $l_{1l}$ = 14 m<br>$l_{2l}$ = 19.5 m<br>$l_{3l}$ = 20.5 m<br>$l_{4l}$ = 24.5 m<br>$l_{1r}$ = 7 m<br>$l_{2r}$ = 14.5 m |
| Elevation of the centering position with the ends of the welded new section for<br>–left end<br>–right end | $h_l$ = −0.1604 m<br>$h_r$= −0.1604 m |
| Coefficients of the polynomial describing the deflection of the axis of the pipeline | $a$ = −4.485 × 10$^{-8}$<br>$b$ = 4.591 × 10$^{-6}$<br>$c$ = 1.102 × 10$^{-4}$<br>$e$ = −1.133 × 10$^{-2}$ |
| Pipeline wall material | Steel 09G2S |
| Density of pipeline wall material | $\rho$ = 7 850 kg/m$^3$ |
| Young's modulus of steel pipeline wall | $E$ = 2 × 10$^{11}$ Pa |
| Pipeline outer diameter | $D$ = 1.02 m |
| Pipeline inner diameter | $d$ = 0.996 m |

The data obtained as a result of the calculation according to the proposed calculation algorithm are presented in Table 4.

**Table 4.** Calculation results of the reaction forces that occur in the hydraulic cylinders of devices when fixing the ends of the pipeline, and the forces required for their centering.

| Parameter | Value | |
|---|---|---|
| | Left Side of the Pipeline | Right Side of the Pipeline |
| $X_1c$, N | 60,840.659 | −228,037.316 |
| $X_2c$, N | 501,900.494 | 199,519.112 |
| $X_3c$, N | −885,059.581 | – |
| $X_4c$, N | 400,649.306 | – |
| $P_2$, N | −385,756.233 | −356,149.835 |
| $P_2$, N | 277,925.232 | 170,705.197 |

Taking into account the given initial data, the finite element model of the height position of the investigated pipeline after cutting it with fixing the position of its ends is shown in Figure 20. In this case, the value of the maximum displacement of the ends of the pipeline is 0.05 mm.

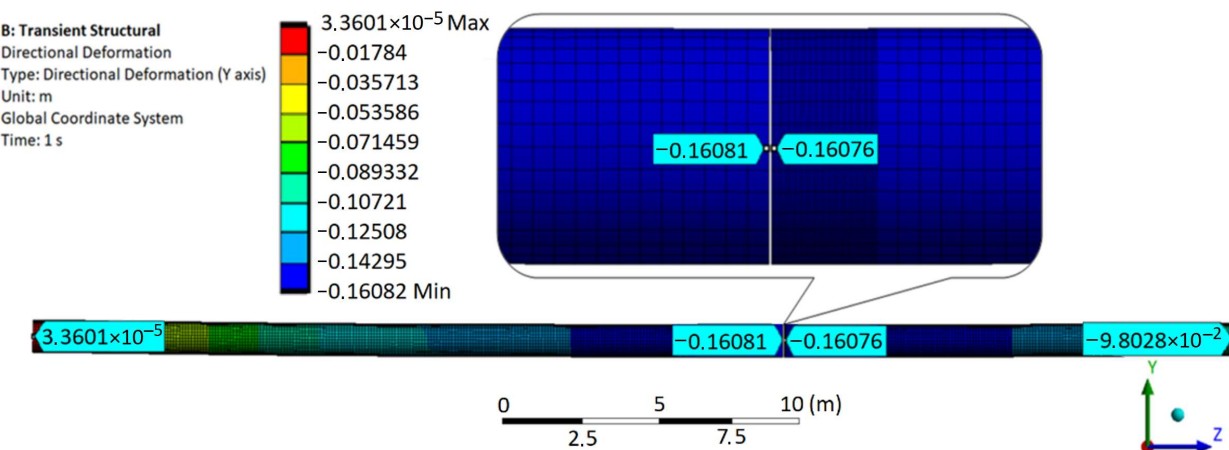

**Figure 20.** The finite element model of the height position of the pipeline after cutting it while fixing the position of its ends.

To further determine the convergence of the forces required to center the ends of the pipeline, we apply the forces $P_1$ and $P_2$, calculated according to the proposed mathematical model to the pipeline. As a result, the height of the ends of the pipeline are $h_l = -0.1556$ m and $h_r = -0.1579$ m.

The finite element model of the height position of the pipeline after cutting out the defective section and centering its ends is shown in Figure 21.

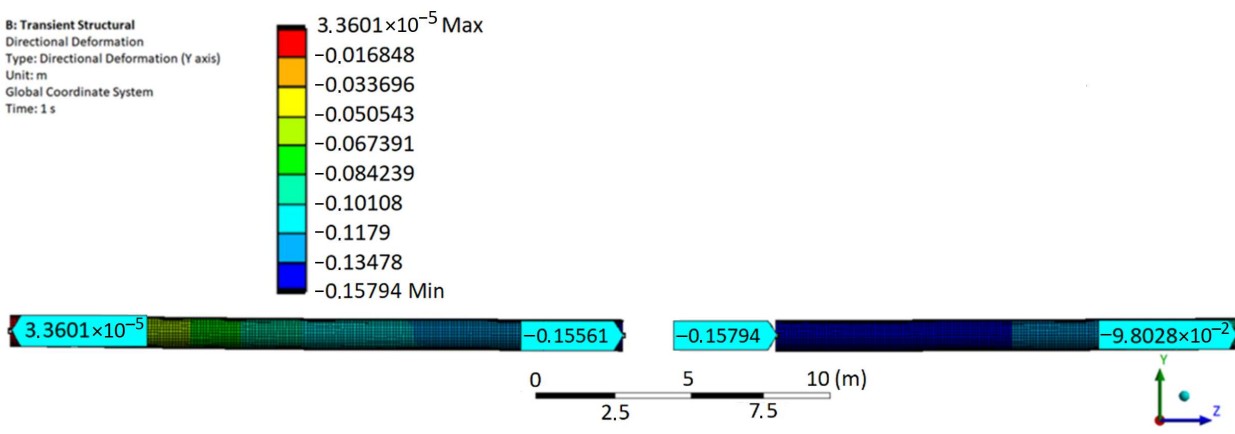

**Figure 21.** The finite element model of the height position of the pipeline after cutting out the defective section and centering its ends.

Relative deviations of the desired values obtained from the results of finite element modeling and using the proposed mathematical model are presented in Table 5.

As a result, the obtained values of $X_{1c}$, $X_{2c}$, $X_{3c}$ and $X_{4c}$, as well as the values of the height of the ends of the pipeline, $h_l$ and $h_r$, when the calculated centering forces are applied to them, differ from those calculated earlier by the proposed mathematical model by less than 5%. The foregoing indicates the correctness of the proposed mathematical model.

### 3.5. Calculation of Financial Costs for Repair Work

To justify the economic efficiency of using the proposed repair method, it is necessary to determine the cost of repair work using the existing repair technology.

The initial data for calculating the annual financial costs for repair work using pipelayers are presented in Table 6.

**Table 5.** The results of the calculation of the reaction forces arising in the hydraulic cylinders of the devices when fixing the position of the modeled pipeline, and the forces required to center its ends.

| Parameter | The Value According to the Proposed Mathematical Model | The Value According to the Data Obtained from ANSYS Software | Relative Deviation of Values, % |
|---|---|---|---|
| $X_{1lc}$, N | 60,840.659 | 58,541.161 | 3.780 |
| $X_{2lc}$, N | 501,900.494 | 492,306.149 | 1.912 |
| $X_{3lc}$, N | −885,059.581 | −862,500.279 | 2.549 |
| $X_{4lc}$, N | 400,649.306 | 381,167.958 | 4.862 |
| $X_{1rc}$, N | −228,037.316 | −219,326.955 | 3.820 |
| $X_{2rc}$, N | 199,519.112 | 193,197.162 | 3.169 |
| $h_l$, m | −0.1604 | −0.1556 | 2.986 |
| $h_r$, m | −0.1604 | −0.1579 | 1.534 |
| $\sigma_{maxl}$, MPa | 186.318 | 181.638 | 2.512 |
| $\sigma_{maxr}$, MPa | 133.017 | 128.541 | 3.365 |

**Table 6.** Initial data for calculating the annual financial costs for repair work using pipelayers.

| Parameter | Value |
|---|---|
| Number of repairs per year carried out by one service department | 40 |
| Pipelayer cost | 385,000 $ |
| Number of pipelayers | 2 |
| Pipelayer service life | 20 years |
| Number of employees in the repair crew | 18 |
| Average salary of an employee | 5 $/h |
| The duration of work on the centering of the ends of the pipeline and their welding | 8 h |
| The cost of a transportation permit for one pipelayer | 800 $/1 repair |
| Diesel fuel cost | 0.9 $/L |
| Pipelayer fuel consumption | 52.1 L/h |
| Truck crane fuel consumption | 35 L/100 km |
| Fuel consumption of the carrier | 35 L/100 km |
| Distance to the place of repair work | 100 km |
| Equipment repair costs (% of equipment cost) | 3% |

The results of the calculation of financial costs are shown in Figure 22.

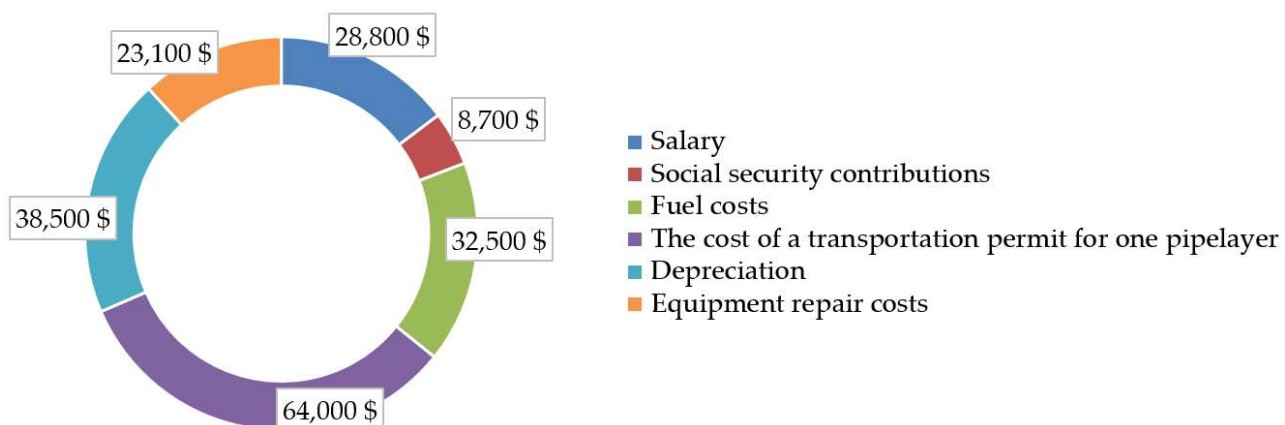

**Figure 22.** The results of the calculation of annual financial costs for repair work using pipelayers.

According to the results of the calculation, it can be seen that the main cost items (transportation permit for one pipelayer, depreciation and fuel costs) fall on the pipelayers used in the repair process. Thus, the replacement of pipelayers with devices for fixing and centering the ends of the pipeline will significantly reduce the cost of repairs. This is ensured by the simplicity of the devices in operation,

their lower capital intensity and the absence of the need to pay for a transportation permit for the oversized equipment and additional fuel costs.

## 4. Discussion

A sharp displacement of the ends of the pipeline when it is cut during the repair process with replacement of the defective section is an urgent problem. The reason for this displacement is the elastic bending of the main pipeline. It should be noted that the pipeline located in the repair trench is most often in an elastically bent state. At the same time, no measures are provided to eliminate the sharp displacement. It is also important to note that, for further centering of the pipeline ends, pipelayers are used, which allow centering only by raising the pipeline ends. To ensure such centering conditions, it is often necessary to carry out additional earthworks. In addition, the transportation of pipelayers requires the payment of permits for the transport of oversized equipment, which in annual terms can reach $100,000 for one service division of the company.

The solution to the above problems is the method of repair of main pipelines developed within the scope of this study. This repair method improves the efficiency of the repair process, as well as the level of industrial safety. This is achieved by using the developed fixing and centering devices during the repair process, which allow fixing of the ends of the pipeline before cutting it to eliminate their sharp displacement, as well as their further centering before welding a new section.

The developed mathematical model makes it possible to carry out a preliminary assessment of the reaction forces arising in the hydraulic cylinders of devices when fixing the ends of the pipeline, the efforts of their centering as well as the stresses arising, in this case, in the body of the pipeline. The mathematical model showed a sufficient level of convergence with the results of finite element modeling of the main pipeline in the ANSYS software package. At the same time, the assessment of the reaction forces arising in the hydraulic cylinders of the devices was carried out on the basis that the oscillations of the pipeline are undamped, which simplifies the mathematical model and, as a result, gives somewhat overestimated values of the reaction forces compared to the actual ones. This excess of the calculated value over the actual value is taken as a safety factor, which makes it possible to ensure greater reliability of the operation of the hydraulic cylinders of the devices during their selection. To clarify the value of this safety factor, in the future it is planned to carry out experimental studies to measure the reaction forces that occur when the pipeline ends are displaced in fixed supports in selected sections of the pipeline.

The initial data for carrying out calculations according to the proposed mathematical model are data on the spatial position of the pipeline in the trench. As a method for determining the spatial position of the pipeline, laser scanning is proposed, the result of which is a spatial point cloud of the pipeline. The developed method of pipeline point cloud processing allows obtaining of a polynomial describing the position of the central axis of the pipeline in space. According to the results of experimental studies, the polynomial obtained as a result of applying this method in comparison with the polynomial obtained as a result of photographing the cross markers glued along the side generatrix of the pipeline allows, with an accuracy of 5%, the assessment of the studied values of the reaction forces, centering forces and the resulting stresses in the pipeline wall.

The developed mathematical model makes it possible to carry out calculations for pipelines with different diameters, wall thicknesses as well as wall materials with different strength parameters. The diameter and wall thickness of the pipeline are taken into account when calculating its rigidity and axial moment of inertia. As for the strength parameters of the pipeline, its plasticity plays the most important role here. The plasticity of the pipeline during calculations affects the allowable stress values that can occur in its wall without plastic deformation during repairs. Based on this, the yield strength of steel is taken as allowable stress value so that the pipeline deformations occur in the elastic zone. This value is taken into account for a specific pipeline material during the calculations made using the proposed mathematical model.

Nevertheless, it should be noted that during the operation of the pipeline, the material of its wall ages. The aging process of the pipeline wall material consists in changing its strength properties [59]. This, in particular, affects the yield strength and tensile strength of the material and may be especially relevant for pipelines with a long period of operation. Of particular interest for this study is the change in the yield strength of the pipeline wall material during its operation. This parameter in the process of steel aging increases to a peak value, and then decreases to a value 10% lower than the initial one [60]. In order to take into account such a change in properties, it makes sense to reduce the allowable values of stresses in the pipeline wall used in calculations according to the proposed mathematical model by a safety factor equal to 10–15%. This will ensure the possibility of centering the ends of the pipeline in the elastic zone of its deformations, even for long-term operated pipelines.

As for possible defects in the pipeline wall, they do not have a significant effect on the calculation results, since the process of centering the ends of the pipeline is carried out after cutting out its defective section. That is, in this case, there are no stress concentrators in the pipeline wall, which cause uneven distribution of stresses.

It is also worth noting that the main types of stresses that occur in the pipeline wall are bending stresses. In the process of centering the ends of the pipeline, possible tensile stresses caused by longitudinal elongation or contraction of the pipeline are eliminated through the use of gripper rollers in the design of the proposed devices. Due to this, the grips of the devices do not create significant longitudinal deformations of the pipeline wall, but only its bending is carried out due to transverse movements of the pipeline ends.

## 5. Conclusions

The following conclusions were made:

1. The disadvantages of the existing methods of repairing main oil and gas pipelines involving cutting out defective sections are justified.
2. The design of devices for fixing and centering the ends of the pipeline has been developed and patented, which makes it possible to avoid a sharp displacement of the ends of the pipeline and ensure their centering before welding a new section without the use of pipelayers.
3. A mathematical model has been developed for calculating the reaction forces arising in the hydraulic cylinders of the fixing and centering devices, the forces required to center the ends of the pipeline as well as the resulting stresses in the pipeline wall. The initial data for calculations based on the mathematical model are data on the position of the pipeline in the repair trench. The mathematical model will ensure the safe automatic operation of devices without the presence of working personnel in the trench.
4. The choice of the method of laser scanning of the pipeline to determine the position of the pipeline in the repair trench is justified.
5. A method for processing a point cloud of a pipeline obtained as a result of laser scanning has been developed. The method allows us to obtain a polynomial that describes the spatial position of the central axis of the pipeline, based on its point cloud.
6. As a result of a laboratory study, it was found that the developed method for processing a point cloud of a scanned pipeline makes it possible to obtain a polynomial, on the basis of which it is possible to estimate with sufficient accuracy (up to 5%) the parameters calculated by the mathematical model, in comparison with the method of photographing cross markers glued on the side generatrix of the pipeline.
7. As a result of modeling the process of cutting the main pipeline and centering its ends by the method of finite element computer simulation, a sufficient convergence (up to 5%) of the simulation results with the results of calculations using the proposed mathematical model was obtained.
8. According to the results of calculating the financial costs for the production of repair work according to the currently used method, it was revealed that the main costs (transportation permit for one pipelayer, depreciation and fuel costs) fall on the use of pipelayers. The devices proposed in the work can significantly reduce these cost items and completely eliminate the largest one, which is the cost of a transportation permit for oversized equipment.
9. The developed method of repairing main oil and gas pipelines and the mathematical model will improve the level of industrial safety of the repair process, as well as its technological and economic efficiency.

## 6. Patents

The results of this study are contained in the patent «Device for fixing and centering the ends of the pipeline when cutting out its defective section», No. 2763096, at the Federal Institute of Intellectual Property of the Russian Federation (registration date 27 December 2021).

**Author Contributions:** Conceptualization, I.S., E.D. and D.S.; methodology, I.S. and E.D.; software, E.D.; validation, I.S. and D.S.; formal analysis, E.D.; investigation, E.D.; resources, D.S.; data curation, I.S.; writing—original draft preparation, E.D.; writing—review and editing, E.D. and I.S.; visualization, E.D. and D.S.; supervision, I.S.; project administration, I.S. All authors have read and agreed to the published version of the manuscript.

**Funding:** This research received no external funding.

**Institutional Review Board Statement:** Not applicable.

**Informed Consent Statement:** Not applicable.

**Conflicts of Interest:** The authors declare no conflict of interest.

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
