# Peer review of "Improving the Method of Replacing the Defective Sections of Main Oil and Gas Pipelines Using Laser Scanning Data"

_applsci, doi:10.3390/app13010048_

Round 1

Reviewer 1 Report

The study has provided a new insights into a novel approach to pipeline repair through deploying laser technique, which is very sound and comprehensive study, underpinned by its robust methodology and findings. The material and method section; and results can benefit from using a flow chart to show the sequence of the lengthy steps followed , to enable readers navigate easily and benefit from the study, and again will improve the communication of findings.

Author Response

Thank you for your informative review!
In accordance with your recommendations, I made the following additions to the text of the article:
1. A flowchart has been added describing the process of repair work.

In addition:

2. Variables are described that affect the accuracy of the calculation results for the proposed mathematical model.
3. The effect of a change in the strength properties, in particular, the ductility of the steel of the pipeline wall, on the calculation results is described.
4. The financial costs were calculated for the repair work according to the currently used method with the use of pipelayers. The economic efficiency of the proposed repair method is substantiated.
5. The nomenclature was provided.

Reviewer 2 Report

Comments are included in pdf file

Author Response

Thank you for your informative review!
In accordance with your recommendations, I made the following additions to the text of the article:

1. Variables are described that affect the accuracy of the calculation results for the proposed mathematical model.
2. The effect of a change in the strength properties, in particular, the ductility of the steel of the pipeline wall, on the calculation results is described.
3. The financial costs were calculated for the repair work according to the currently used method with the use of pipelayers. The economic efficiency of the proposed repair method is substantiated.
4. A flowchart has been added describing the process of repair work.

In addition:

5. The nomenclature was provided.

Reviewer 3 Report

See the reviewer's comments file.

Author Response

Thank you for your informative review!
In accordance with your recommendations, I made the following additions to the text of the article:
1. Studies have been added from articles on the mechanics of the flow of fluids in the pipeline, as well as the influence of the composition of oils on the operation of the pipeline.
2. Figures 2-4 have been reconstructed and presented in better quality.
3. Added links to sources when mentioning equations.
4. The nomenclature was provided.

5. The left part of Fig. 15 have been revised and presented in better quality.

In addition:

6. Variables are described that affect the accuracy of the calculation results for the proposed mathematical model.
7. The effect of a change in the strength properties, in particular, the ductility of the steel of the pipeline wall, on the calculation results is described.
8. The financial costs were calculated for the repair work according to the currently used method with the use of pipelayers. The economic efficiency of the proposed repair method is substantiated.
9. Added a flowchart describing the process of repair work.